# Compositions Based on PAN Solutions Containing Polydimethylsiloxane Additives: Morphology, Rheology, and Fiber Spinning

**DOI:** 10.3390/polym12040815

**Published:** 2020-04-04

**Authors:** Valery G. Kulichikhin, Ivan Y. Skvortsov, Lydia A. Varfolomeeva

**Affiliations:** A.V. Topchiev Institute of Petrochemical Synthesis, Russian Academy of Sciences 29, Leninsky prospect, Moscow 119991, Russia; amber5@yandex.ru (I.Y.S.); varfolomeeva.lidia@mail.ru (L.A.V.)

**Keywords:** rheology, solution, emulsion, polyorganosiloxanes, polyacrylonitrile, fiber spinning

## Abstract

The effect of additives of polydimethylsiloxanes (PDMS) with various molecular weights on the morphology and rheological behavior of polyacrylonitrile (PAN) solutions in dimethyl sulfoxide has been analyzed. It was shown that only partial compatibility of the PDMS with the lowest molecular weight member of the homologous series studied—hexamethyldisiloxane—with PAN solution takes place. All other PDMS samples form emulsions with PAN solutions. The coalescence rate of PDMS drops depends on the viscosity ratio of the disperse phase and the continuous medium, which determines both the duration of dispersion preparation and the conditions for processing emulsions into fibers and films. An anomalous change in viscosity for a series of emulsions with different concentrations of additives, associated with the slippage, was detected. The relaxation properties of emulsions “feel” macro-phase separation. Modeling of the wet spinning process has shown that the morphology of the deposited solution drop reflects the movement of the diffusion front, leading to the gathering droplets in the center of the deposited formulation drop or to their localization in a certain arrangement. It was shown that the emulsion jets upon stretching undergo phase separation.

## 1. Introduction

Introducing organosilicon compounds into polymer melts and solutions is a rather common direction in modern polymer science and technology, as described in reviews (see, for example, [1]). As a rule, this approach is stipulated by wishing to combine, in one product, carbon chain polymers and silicate fragments that in some cases are important to reach the desired functional properties of the hybrid materials in practice [2]. This approach is especially interesting for spinning fibers—precursors of carbon fibers. The combination of carbon and silica carbide or oxycarbide structures a priori should introduce into the properties of the final fibers the needed level of hydrophobicity and control of the thermo-deformation characteristics. 

For such dopes, it is difficult to achieve the total compatibility of components, which is why it should be expected that the definite level of dispersity in solutions, as in final fibers, may present. In addition, their important peculiarity is the presence in their compositions of one or more polymer phases, which means to bear in mind the necessity of the detailed analysis of not only the rheological properties but also the relaxation properties typical for macromolecular chains.

There exist different methods of introducing organosilicon compounds into polymer matrices. This could be polymer synthesis in the presence of such additives [3], the sol-gel processes [4,5], the mechanical mixing in solutions or melts [6,7], and the chemical grafting of silicon-containing groups to macromolecules [8,9,10]. For preparing hybrid composite films and fibers with included silicon carbide [11,12], the most preferable host polymers are precursors of the carbon fibers—polyacrylonitrile (PAN) and cellulose. Several papers devoted to hybrid fibers by means of addition to the matrix polymer of tetraethoxysilane (TEOS) [7] and vinyltriethoxysilane [13] into dopes with the subsequent carbonization of fibers-precursors were published. At thermal treatment, silanes transform to silicon carbide particles distributed in a matrix of the carbon fiber.

PAN is the most popular polymer for preparing the high-tenacity carbon fibers due to specific structural transformations at thermolysis, resulting in high carbon yields reaching 40% [14,15]. One of the best solvents for PAN is dimethylsulfoxide (DMSO) [16], partially dissolving TEOS, and this capability of the solvent allows the preparing of mutual solutions, as emulsions of the various composition [4]. 

An interest to the addition of polydimethylsiloxanes (PDMS) to PAN and other polymers can be explained by the unique features of silicon organic polymers, namely hydrophobicity, high chain flexibility, and capability of transformation upon heat treatment to cyclic siloxanes, as well as to silicon oxycarbide and silicon carbide in the presence of carbon [17]. The low glass transition point, a wide range of molecular weights of produced PDMS, and high thermal stability lead to considering them as important components for use as additives in hybrid fibers [6,12,18], membranes—including composite membranes of different morphology [19,20]—and so-called “breathing” membranes with decreased water uptake [21]. They also can be used for the modification of hollow fibers [19,22,23] and functionalization of monolith fibers by coating with mixtures of PDMS with polypyrrole—for realizing the electrical conductivity [24,25]. The member of the PDMS family most compatible with PAN is hexamethyldisiloxane (HMDSO)—one of the well-known silicon organic oligomers. There are indications [26] of its introduction into PAN as a component, reinforcing the effect of plasma treatment for improving the surface characteristics of fibers. 

The mixtures of incompatible PDMS solutions in acetone and PAN in dimethylformamide were studied in [12]. From the obtained emulsions, the hybrid fibers were prepared by means of electrospinning, and their structure and thermal properties were investigated. It was proved that Si-C links form after heat treatment to 1000 °C. The fibers have a skin-core morphology with a decreased silicon content in the skin compared with the average in the fiber volume. A similar effect of non-homogeneous silicon distribution at the model coagulation of a drop of the PAN solution in DMSO with TEOS additives by aqueous coagulants was observed in [4].

As a rule, an increase in molecular weight leads to a decrease in polymer solubility [27,28]. That is why in the present paper, the influence of PDMS—at a wide range of molecular weights—on the morphology of the dopes (solutions and emulsions), as well as on their evolution at interaction with coagulants, their phase behavior at stretching homogeneous and heterogeneous jets, and their rheological properties were analyzed. In the latter case, the main characteristics of the analysis were the viscosity of the disperse phase, that could be either higher or lower than that of the matrix PAN solution, and the relaxation properties. As a whole, this research was devoted to the development of spinning regimes of hybrid fibers and films with controlled morphologies and properties. 

## 2. Materials and Methods

### 2.1. Materials 

Solutions of the ternary PAN copolymer (Good Fellow Co, Huntingdon, Great Britain)—containing 93.8% of acrylonitrile, 5.8% of methylacrylate and 0.3% of methylsulfonate—with a molecular weight of 85 kg/mole and a polydispersity index of 2.1 in DMSO (ECOS-1, Moscow, Russia) have been studied. As silicon organic components, linear PDMS of different molecular weights—HMDSO (ECOS-1, Moscow, Russia), polydimethylsiloxane-5 (PDMS-5, Solins, Dankov, Russia), polydimethylsiloxane-100 (PDMS-100, Silane, Dankov, Russia), polydimethylsiloxane-400 (PDMS-400, Silane, Dankov, Russia), and polydimethylsiloxane rubber (SKTN-E, Silane, Dankov, Russia)—were used. All of these objects were liquids and, depending on molecular weight, demonstrated either Newtonian or viscoelastic behavior. Their characteristics are presented in Table 1.

Preliminarily, under agitation for four hours at 70 °C, the 20% matrix solution of PAN in DMSO was prepared. For the matrix solution, a stirrer with a J-like rotor and a 60 rpm speed (Heidolph RZR-2020, Schwabach, Germany) was used. The same stirrer with a speed of 600 rpm was applied for emulsions. Then, at a temperature of 50 °C and with agitation for one hour, the compositions based on matrix PAN solution and HMDSO (0.1–20%), PDMS-5 (0.1–30%), PDMS-100 (0.1–5%), PDMS-400 (0.1–30%), and SKTN-E (0.1%–5%) were obtained. They were tested by optical and rheological methods.

For the investigation of composition behavior at extension deformation, the matrix 25% solution and corresponding mixtures with PDMS were used.

An approach to the real regimes of the wet spinning was preliminarily chosen in model conditions—by means of the optical analysis of the diffusion interactions of the composition drop, modeling the cross-section of the jet/fiber surrounded with coagulant. In all cases, as the coagulant, a mixture of water and DMSO at a 15:85 ratio was used.

### 2.2. Methods

#### 2.2.1. Rotational Rheometry

The rheometer ThermoHaake RheoStress RS600 (Thermo Fisher Scientific Inc., Waltham, MA, USA), with an operating unit cone and plate with a cone diameter of 60 mm and an angle of one degree, was used for rheological measurements. In the steady-state mode of strain, the flow curves were obtained in a shear rate range of 10^−1^–10^4^ s^−1^. For determining the domain of linear viscoelasticity, the complex modulus of elasticity and its components—the storage G’ and loss moduli G” in the strain range of 0.01%–100% at constant frequencies of 6 and 500 rad/s—were measured. The frequency dependences of both moduli in the linear domain of viscoelasticity were measured in a frequency range of 0.6–628 rad/s. All measurements were performed at 25 °C.

#### 2.2.2. Optical Microscopy

The polarization microscope Biomed 6 PO (Biomed, Moscow, Russia) for optical observations of the composition morphology was used. For this purpose, a drop of solution was placed between the slide and cover glasses at a gap thickness of ~100 μm. Images were obtained by the microscope camera ToupTek E3ISPM5000 (ToupTek Photonics Co, Hangzhou, China), with a resolution of six dots per micron.

#### 2.2.3. Modeling of the Coagulation Process

For the estimation of the effects of the introduced additives on the coagulation of the PAN solution caused by the action of the coagulant, a drop of composite solution was placed between the slide and cover glasses and was surrounded with coagulant. For the exclusion of the interaction of the drop with the moisture of air, this process was realized in a dry box at relative humidity of less 1%. Within 10 min from beginning the contact between the solution drop and coagulant, the morphology of the deposited drop was analyzed by optical microscopy in transmitted light.

In addition, for more detail research of the coagulation kinetics and evolution of the diffusion zone on the border between the solution and coagulant, a method of optical interferometry was used [4]. This method was sufficiently informative, not only for the analysis of the phase separation of PAN solution under the action of coagulant, but also for the location of droplets of silicon organic phase in the course of PAN solution coagulation.

#### 2.2.4. Modeling of Mechanotropic Spinning

The experiment involves stretching a solution drop produced by a syringe and transforming the drop into the jet at different speeds (Figure 1). The most original element of this device is the system of lighting. The light beam from a halogen lamp (150 W) passes through an optical fiber and is focused in the center of a drop by means of a microscope objective. In other words, a stretching jet itself plays the role of optical fiber. In addition, backlighting, to make clear the boundaries of a jet/fiber, was applied. Videography started simultaneously with the moving of a needle with a drop of solution. The resolution was 1920 × 1080 with a frequency of 60 shots per second. The objective used (produced by LOMO, St-Petersburg, Russia) provides the necessary clarity and depth of vision.

The experiments were carried out according to the following scheme:A drop of a solution with a volume of 10 μL was squeezed out of a syringe located at the bottom of the unit.By moving the syringe, a drop was brought into contact with the upper plate of the optical fiber lens.The droplet was stretched at a constant speed of 0.65 mm/min by moving the syringe to a distance of 11 mm.

At the initial moment of the droplet stretching, the video recording process was started at a frequency of 60 fps on a Touptek XFCAM1080PHD camera (ToupTek Photonics Co, Hangzhou, China), coupled to a LOMO 4x lens, which allowed obtaining images with a resolution of 5.5 μm.

## 3. Results and Discussion

### 3.1. Morphology

Low molecular weight oligodimethylsiloxanes are hydrophobic, nonpolar liquids with a very low energy of dispersion interaction [28], which significantly limits their solubility in polar aprotic solvents and, especially, in solutions of polar polymers. A study of the morphology of the obtained mixed systems showed that, depending on the molecular weight of polyorganosiloxane, the formation of three types of emulsion is possible.

**Emulsions of the First Type** HMDSO is soluble in solutions of PAN in DMSO to a concentration of 1%. Mixtures in the concentration range of HMDSO from 2% to 5% are emulsions with a droplet size of the dispersed phase of the order of 2–4 microns (Figure 2), stable two days. With increasing concentrations above 5%, emulsions become unstable due to the low viscosity of HMDSO (0.5 Pa·s) and poor affinity for PAN solutions in DMSO, which is manifested in the coalescence of drops of HMDSO and their accumulation on the surface of the mixture. This effect can be explained by the flotation process of HMDSO drops due to the difference in densities with the PAN solution.

In this case, the droplet shape remains spherical and droplets have almost the same size distribution regardless of the concentration of the additive (Figure 3).

The flotation effect also occurs for emulsions of PAN solution with PDMS-5. This leads to a sharp gradient in the content of the dispersed phase droplets in the bulk and on the surface. Figure 4 shows microphotographs of PAN solutions with various amounts of PDMS-5 in a flat cuvette. With an increase in the concentration of PDMS, the number of drops first increases, and then sharply decreases. The fact is that, at a concentration above the critical level, intense coalescence of microdroplets occurs, with the formation of large droplets that spill onto the surface of the solution. In other words, the macro-stratification of a freshly prepared emulsion takes place, and large droplets are released from the volume of the solution onto the surface, coalescing with the surrounding microdroplets of the dispersed phase, causing a decrease in their concentration in the volume.

**Emulsions of the second type** are formed when higher molecular weight polyorganosiloxanes, namely PDMS-400, but with a lower viscosity than the viscosity of the PAN solution are added. Over the entire range of concentrations, emulsions are formed with a droplet size of 2–20 μm (Figure 5b), and the distance between the droplets is significantly higher compared with emulsions with HMDSO (Figure 5a). Emulsions with PDMS-5 are stable for more than three days, and with PDMS-100 for two weeks. 

**Emulsions of the third type**, namely multiple emulsions, are formed when polysiloxanes with a viscosity higher than that of the matrix solution are added into solution. Systems with 5% polyorganosiloxane are a mixture of conventional and double emulsions with droplet sizes of 2–4 and 20–70 μm, respectively, and the average diameter of the inner drops of the PAN solution in SKTN-E drops is in the order of 3–5 μm. Such emulsions preserve stability for more than one year. A micrograph of the PAN solution with SKTN-E is shown in Figure 5c. 

The preparation of samples for microscopic viewing based on a solution of PAN and highly viscous PDMS-400 and SKTN-E with their content above 5%, by compressing the resulting thin layer with a cover glass, leads to three important effects. The first consists of the appearance of extended sequences of droplets forming rings relative to the center of the droplet (Figure 6). The second effect consists of a concentration gradient of droplets of the dispersed phase along the radius of the droplet. The periphery is more saturated with drops of PDMS-400 than the middle of the sample. Finally, judging by the contrast of the periphery and the central parts of the preparation, it is possible to suppose that compression and biaxial tension induce phase inversion: an emulsion with an polyorganosiloxane dispersed phase is retained inside the droplet (A), and a PAN solution already acts as the dispersed phase on the periphery (B).

The reasons for such a variety of morphologies of a single flattened droplet of the composition are not yet clear, but as a possible explanation, we can hypothesize the role of the elastic strain developed during the compression of an emulsion with a viscoelastic dispersion medium and a dispersed phase. This hypothesis will be formulated in more detail in the analysis of the rheological properties of the compositions.

Taking into account the large number of compositions under investigation before rheological testing, their compositions, morphologies, and stabilities are collected in Table 2. 

### 3.2. Rheology

Before conducting rheological experiments, it is necessary to ensure the stability of the heterogeneous objects under consideration. Based on visual and optical observations, we estimate so-called static stability in time to perform measurements inside the period of static stability. In the work with emulsions having rather mobile structures, it is important to standardize the preparation of the specimens under equal conditions. The issue is that at the loading into operating unit and the squeezing of the emulsion layer between the cone and plate, some disturbances of the initial structure are induced. For preliminary conditioning, the following procedure, tested on emulsion containing 5% of PDMS-100 in a 20% PAN solution, was performed: surveillance of the evolution of viscosity in time (Figure 7). 

Three shear rates were applied. As is seen, the most indicative is the low shear rate, while for the highest rate, an initial strong decrease of viscosity is observed. We need to keep in mind that for heterogeneous systems, the start-up shearing causes a change in the emulsion structure, but then viscosity values become constant in time, corresponding to measured values in the traditional sweep of rates. In addition, according to [32], the following protocol of emulsion conditioning was preliminarily tested on a chosen dispersion: 30 s shearing with a rate of 50 s^−1^ and storing for 2 min at 20 °C. The reproducibility of the results for steady state and oscillatory shearing for at least five experiments and the coincidence with usual shear rates sweep was tested and successfully proven. The following flow curves for 20% PAN solution containing HMDSO, PDMS-5, PDMS-100, PDMS-400, and SKTN-E in a sweep regime, taking into account the start-up time, were obtained (Figure 8A–E).

All systems under consideration, with the exception of highly concentrated emulsions containing PDMS-100 and PDMS-400 with a concentration above 10%, exhibit a non-Newtonian character of flow. For systems with PDMS-100 and PDMS-400, starting from a 10% concentration, a sharp decrease in viscosity is observed at low shear rates, which may be due to the manifestation of viscoplastic behavior or slip effects (Figure 8) [33]. A similar suggestion can be made about the sharp decrease in viscosity at high shear rates. On these branches of the flow curves, the difference for systems containing different amount of PDMS is very likely be explained by not only “interphase”, but also “instrumental” slippage of the compositions relative to the measuring walls of the operating unit. This behavior is associated with the concept of the “spurt” effect as a result of the forced transition of the surface layers of the measured system to a rubber-like state under the influence of an intense mechanical field [34].

Features of the rheological behavior of systems containing PDMS-100 and PDMS-400 are also manifested in the concentration dependences of viscosity, presented in Figure 9.

If the introduction into solution of compatible up to concentrations of 1% HMDSO leads to a decrease in viscosity, which is due to dilution of the system with a low-viscosity component, the further stabilization of viscosity requires a special explanation. It is possible that a low-viscosity unstable emulsion is separated into the phases of the solution and HMDSO, with the excess of the latter emitted to the surface as a result of flotation, and the concentration of the additive in the volume remaining constant. The obtained data correlate with the morphology of the emulsions presented in Figure 3.

An increase in the molecular weight of polyorganosiloxanes leads to a loss of its compatibility with PAN solutions, but the stability of emulsions increases substantially in proportion to the viscosity of the additive. The other factors affecting the rheology of emulsions are the structure and dimensions of the interfaces. It is understandable that we cannot not perform the direct measurements of the interface structure and dimensions. The above-mentioned assumption is initiated by two sets of indirect data: the morphology of the emulsions (Figure 4 and Figure 5) and the concentration dependences of the viscosities (Figure 9). These factors most clearly work for emulsions with PDMS-5. The presence of droplets of polyorganosiloxane liquid in the system causes an increase in and subsequent stabilization of viscosity at all concentrations studied. If the stage of viscosity growth can be explained by the formation of a “network” of interfacial boundaries, then the stabilization can be explained by the same reasons as in the case of HMDSO, i.e., the flotation of large droplets to the periphery of the stream and maintenance of constant the residual composition of emulsions in the volume. 

The addition to PAN solution of more viscous PDMS-100 and PDMS-400 causes a maximum in the range of additive concentrations of 2%–5% and a subsequent decrease in viscosity. The increase in viscosity may be due to the presence in the system of a component that is more viscous than the matrix solution, but the subsequent decrease is most likely caused by interfacial slippage and a shear-induced redistribution of droplets of the dispersed phase between the central part and the periphery of the stream (see Figure 6). This can cause the appearance of a yield strength, the signs of which are visible from the flow curves of these compositions. Finally, in the case of SKTN-E, the double emulsion forms, that increases the interface density, and this effect causes strong viscosity growth along the concentration axis. In some cases, the morphology of emulsions is changed drastically with the changing concentration of the disperse phase due to coalescence, flotation, etc. In addition, we did not consider as whole an influence, on viscosity, of the strength and the elasticity of interfaces. This factor was also not measured, though it could be estimated by visualization of the shape of the disperse phase droplets at flow, but in the frame of this paper, this kind of experiment was not performed.

Coming to the discussion of the results of dynamic measurements, it should be noted that in the presence of polyoligosiloxanes, regardless of the type and concentration, there is virtually no effect on the loss modulus of the compositions, while the values of the elastic modulus vary slightly depending on the concentration of the PDMS and its molecular weights (Figure 10). However, these changes at the concentration scale are so small that they do not allow the reliable judging of structural changes in the compositions.

The relaxation properties of the compositions turned out to be more informative. Earlier [4], we evaluated the transition effect of PAN solution containing TEOS in emulsion by analyzing changes in the intrinsic relaxation time. This approach made it possible to record the maximum relaxation time in the region of phase separation and the appearance of interphase boundaries. In this work, a similar behavior could be expected only for compositions with HMDSO, while for others, the main phase change in the system is the macro-separation of emulsions into continuous phases. It was of interest to find out whether this process affects the intrinsic relaxation time, i.e., what the features of the evolution of relaxation times in emulsions with PAN solution matrix and PDMS components of various molecular weights are.

Therefore, we consider the corresponding data for the increase in the relaxation time Δλ relatively to the corresponding relaxation characteristic of the neat PAN solution at different frequencies, determined, in accordance with [35], by the equation:(1)$$\lambda =\frac{G^{\prime }}{\left|\eta^{*}\right|\times \omega^{2}\;\;\;\;}$$
where |η*| is the complex viscosity, ω is the angular frequency, and G’ is the storage modulus. Thus, the relative relaxation time (Δλ) was calculated as the ratio of the corresponding times of the mixed system (λ_PAN + DMSO + PDMS_) and the initial solution (λ_PAN + DMSO_) determined at the same frequency.

Figure 11 shows a series of Δλ values at different frequencies for mixtures of a PAN solution with polydimethylsiloxanes. As for partially soluble TEOS, which reduces Δλ due to dilution in the range of solubility concentrations [4], the addition of HMSDO reduces the specific excess relaxation time at a concentration of up to 2%. This is followed by an increase in Δλ due to the formation of an emulsion with a slight maximum at 5% (the appearance of developed interphase boundaries).

For systems containing PDMS-5, the relative relaxation time increases when the additive concentration is 5%, where the developed emulsion is formed, after which the macro-phase separation occurs, and the dispersed phase drops move to the surface, capturing smaller droplets along the way (see Figure 4). As a result, the number of droplets and the fraction of interphase boundaries in the volume decrease, which leads to a decrease in the relative relaxation time.

Systems containing polydimethylsiloxanes of higher molecular weight than PDMS-100, form more stable emulsions, which are characterized by a higher density of interphase boundaries; therefore, with an increase in the concentration of these components, the relaxation time increases. In the case of PDMS-400, after a maximum at 5% of the additive, a decrease in Δλ at 10% and a subsequent increase proceeds. Judging by the flow curves, there is a tendency of this composition toward the appearance of a yield stress (Figure 8) and the presence of a maximum in the concentration dependences of viscosity (Figure 11). It is possible that the minimum Δλ at a concentration of the dispersed phase of ~10% is due to interfacial slippage, and that the subsequent increase in the intensity of the relaxation process is due to partial phase inversion and the implementation of a mixed morphology of the composition. A similar situation can occur for compositions with SKTN-E, for which phase inversion occurs at much lower contents of polyorganosiloxane compounds.

### 3.3. Modeling the Wet Spinning Process

As mentioned above, the introduction of a hydrophobic organosiloxane compound into a spinning dope of PAN and the preparation of composite fibers are most important from the point of view of the subsequent carbonization of such precursor fibers. It is possible that in the process of thermolysis, carbon fibers reinforced with silicon carbide particles can be obtained. However, such studies should be preceded by experiments on obtaining “white” composite fibers, because their morphology and structure determine the quality of carbon fibers. For this reason, some aspects of the spinning of the composite PAN fibers containing a polyorganosiloxane phase are considered. First of all, we were interested in the kinetics of the coagulation of heterophase solutions—in particular, in terms of the nature of the distribution of polydimethylsiloxanes over the fiber cross section. The experiments on a drop, simulating a cross section of a fiber surrounded by a coagulant, were performed.

Data on the coagulation of solutions containing PDMS are presented in Figure 12.

Microscopy with bright-field illumination for the visualization of the dispersion drop surrounded with coagulant was used for obtaining the pictures. The addition of 1% HMDSO does not visually affect the coagulation process—the system remains homogeneous during the transformation of the droplet from a liquid- to gel-like state. With an increase in the concentration of HMDSO to 5%, the hydrophobic additive migrates to the center of the drop, leaving a neat solution at the periphery. In the emulsion moving toward the center, droplets remain separated with a size of 1–4 μm, i.e., a decrease in the distance between them does not lead to intense coalescence. This phenomenon is similar to that observed during the coagulation of PAN solution in DMSO with TEOS, which has limited solubility in an amount higher than 10% [4] and, most likely, should lead to the formation of a skin-core fiber morphology.

When the coagulation of compositions containing PDMS-5 is considered, the redistribution of microdroplets of the dispersed phase along the drop diameter does not occur. Nevertheless, for a system with 5% PDMS-5, droplet coalescence is observed, although the interdiffusion of the coagulant (non-solvent) in and the solvent outside of the drop of the PAN solution proceeds smoothly—defects and inhomogeneities caused by unsteady diffusion are not detected.

The most interesting situation is with the deposition of compositions by moderately viscous additives PDMS-100 and PDMS-400, where there is a redistribution of the local positions of small droplets of a dispersed phase from chaotic to ordered. This means the formation of extended sequences of droplets while maintaining the constancy of their average content. There are several reasons for this ordering. The first is the orientation of the droplets of the dispersed phase under the influence of the diffusion front during the coagulation of the dispersed medium. The second possible reason is the intense mechanical effect on the sample during the application of the emulsion drop from a tip of needle to a glass slide. In this case, part of the drop adheres to the surface of the glass, and a small part is pulled out after the removing the needle. During such application and subsequent compression of the sample, a complex stress state of the droplet (a combination of extension, squeezing, and shear) is realized. It is possible that under such conditions, the elastic deformations of a viscoelastic solution become significant, initiating the appearance of extended sections in the sample with relaxation properties corresponding to almost elastic body. At their boundaries, the dispersed phase droplets are concentrated. Such a mechanism of the ordering in heterophase systems was described in [36].

Apparently, this reason is the main one, since such clusters of droplets appear immediately after sample preparation, and not as a result of the mutual diffusion of the solvent and coagulant over time. Meanwhile, the ring ordering of droplets of the polyorganosiloxane phase occurs for the same system at observation by optical microscopy, accompanied by a partial phase inversion (Figure 6). Apparently, for PDMS-100 and PDMS-400 additives, the optimum ratio of the viscosities of the dispersion medium and the dispersed phase is achieved, which contributes to the implementation, upon tension and compression, of a certain level of elasticity of the matrix PAN solution and the mobility of polyorganosiloxane droplets.

In the case of more viscous SKTN-E, the double emulsion becomes polydisperse with randomly distributed droplets. No involvement of internal drops of the PAN solution in the coagulation process was noticed. In the first approximation, the same applies to the polyorganosiloxane phase, but to verify this conclusion, kinetic studies, using the method of optical micro-interferometry were carried out. The version of the used micro-interferometry method consisted of the observation of interference fringes in contacting along the interface of the emulsion and coagulant. In the case of partial compatibility of interacting media, the interference bands are bending in the vicinity of interface, as a result of the interdiffusion process proceeding in time and causing a change of the refraction indices. 

The kinetics of the coagulation process of systems based on matrix PAN solution containing 5% of HMDSO and PDMS-400 is presented in Figure 13. In mixtures with HMDSO, the interaction of the initial composition with the coagulant is accompanied by a partial dissolution of the additive in the PAN solution and the appearance of an additional phase of the ternary system with the own fringes and specific pitch in the zone between the composition and the coagulant (shown by arrows on interferograms). In this case, the droplets of HMDSO during the deposition process diffuse toward the interphase border and concentrate near it. In the case of polydimethylsiloxanes incompatible with the PAN solution, this phenomenon was not observed, i.e., drops of the second phase were not redistributed in the volume during the interdiffusion of the coagulant and the solvent. Judging by the bending of the interference fringes visible in the transparent coagulant, mass transfer occurred only between the PAN solution in the composition and the coagulant (DMSO/water mixture). Some changes, even in large drops of double emulsions, do not occur.

Thus, by varying the hydrophobic polyorganosiloxane additives with different molecular weights and concentrations, it is possible to obtain composite fibers and films with various sizes of inclusion, both with variable and uniform distributions over the cross section, and—consequently—over the volume of the finished product. Such fibers were spun on a wet spinning stand shown in Figure 14, using a die with 100 holes with a diameter of 80 μm using the same coagulant that was used in experiments to simulate the deposition of a drop of the studied compositions (85% solution of DMSO in water).

During the spinning process, it was possible to achieve a spinbond hood (the ratio of the speed of the yarn on the first roller to the linear velocity of the solution jet flowing out of the die) of ~ 200% and a spinning process speed of 15 m/min. The resulting fibers have a circular cross section and a virtually defect-free surface. They are currently being investigated by various methods, including X-ray diffraction, electron microscopy, and testing the mechanical characteristics.

In addition to the classical method of fibers spinning from polymer solutions using a coagulant, i.e., wet spinning, there is also the so-called dry spinning method, during which the solvent either evaporates from the jets of the solution due to high temperature, or is released on the surface of the spinning filament due to strong stretching. This method was called mechanotropic spinning, and earlier [29], it was successfully applied for spinning fibers from neat PAN solutions. In this paper, it is extended to PAN fibers containing polyorganosiloxane inclusions.

### 3.4. Modeling of Mechanotropic Spinning

Mechanotropic spinning was simulated by stretching a jet of a composite solution to a constant length. The corresponding experimental results are presented in Figure 15. The images show the comparative kinetics of the thinning of the liquid filament and the phase separation process according to the mechanism described in detail in [30,31]. 

The initial diameter of the droplet, depending on the viscosity of the solution, ranged from 350 to 500 microns. In the process of stretching the emulsions, it was noted that the droplets of the dispersed phase are deformed only at the beginning of the stretching, in the cone zone, after which they relax to a spherical shape, which does not change further in the thinning zone.

For a PAN solution, a noticeable change in the shape of the jet begins from the 30th second and is accompanied by the release of solvent to the jet surface. A formulation with 1% HMDSO is as homogeneous as the PAN solution, but the thinning of the droplet and, accordingly, the onset of macro-phase separation proceeds faster and more intensively compared to the solution without additives. The system with 5% HMDSO is an emulsion with droplet sizes from 5 to 150 microns. The speed of thinning of the jet in such a system increases significantly, and the onset of phase separation occurs much earlier than for the neat PAN solution (~ 10 sec.), which is apparently due to the dilution of the system with a low-viscous additive and a loss of stability as a result of a decrease in the elasticity of the jet. For additives more viscous than PDMS-100, the rate of the thinning of the liquid jet slows down due to an increase in viscoelasticity, but the kinetics of the process of solvent release to the surface remain unchanged. This means that additives do not affect the stability of the solid fiber formation upon the stretching the jets of the emulsions under investigation. From each of them, it is possible to obtain solid fibers by a mechanotropic method, i.e., without the use of coagulant.

However, in this case, an additional feature appears, consisting in the delay of the phase separation process after the end of the extension of the solution drop. In these images, for the zero-time moment, the stopping time of the stretched drop was chosen. In other words, the strong extension induces the micro-phase separation process, consisting, in appearance, of bright light scattering inside a jet. However, from the viewpoint of fiber spinning, the most important stage is macro-phase separation, i.e., the diffusion of micro-droplets of DMSO to the periphery of the spinning jet/fiber, the formation of the thin surface film, and its transformation to the separate drops. These stages are shown in Figure 16. 

## 4. Conclusions

The effect of a number of oligodimethylsiloxanes and polydimethylsiloxanes of various molecular weights added into PAN solution in DMSO on the morphology, rheological behavior of composite solutions, and the processes of coagulation and phase separation under extension were studied. It was shown that polyorganosiloxanes, with the exception of small additives of HMDSO, are incompatible with PAN solutions, and as a result, the compositions are emulsions. Three different types of emulsion are possible depending on the molecular weight and concentration of the additive:

- In the case of HMDSO, a homogeneous emulsion is formed with a narrow distribution of micron-sized droplets.

- The use of PDMS with a viscosity lower than the viscosity of the polymer solution results in emulsions with a polydisperse droplet size distribution.

- If rubber-like SKTN-E is introduced into the PAN solution, the viscosity of which exceeds the viscosity of the polymer solution, multiple emulsions are formed.

In the emulsions under consideration, the presences of interphase boundaries and their densities have a decisive influence on the rheological behavior of the compositions, initiating, in some cases, viscoplastic behavior, and, under macro-phase separation conditions, interfacial slip, rather than the true flow of the compositions. During the formation of an emulsion, the appearance and density of interphase boundaries leads to a change in rheological behavior, which is especially pronounced when considering the increase in the intrinsic relaxation time of the compositions compared to the value for the PAN solution, allowing us to determine the most optimal concentration of PDMS in the dope for spinning.

The presence of hydrophobic polydimethylsiloxanes in the PAN solution affects the morphology of the composition, which is realized during the coagulation process. During the coagulation, the droplets of the dispersed phase of HMDSO migrate to the center of the solution drop, which simulates the cross section of the spinning fiber. The droplets of polydimethylsiloxanes with a higher molecular weight during the preparation of the sample form oriented chains of droplets due to the mutual influence of diffusion front movement and a complex stress field during sample preparation. 

Modeling the process of stretching a liquid filament showed that the addition of HMDSO accelerates the thinning of the jet and phase separation. High molecular weight additives do not significantly affect significantly the rate of phase separation of a liquid filament into an oriented polymer and solvent that migrates to its surface.

Thus, in the course of the work, optimal formulations based on solutions of PAN and polydimethylsiloxanes of various molecular weights, showing promise as dopes in the preparation of carbon-silicon carbide fibers precursors, were developed.

## Figures and Tables

**Figure 1 polymers-12-00815-f001:**
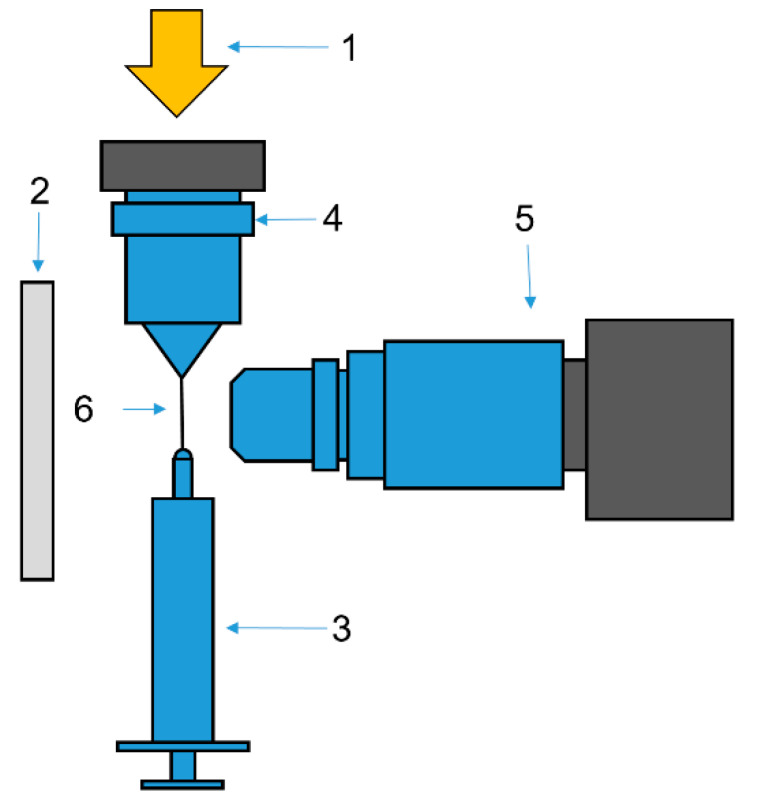
An experimental device for stretching a jet up to a definite length. **1**—a fiber-optic illuminator along the jet axis; **2**—back lighting; **3**—a syringe with the solution; **4**—a lens to focus the light into the center of an extended jet; **5**—a camera; **6**—a jet. This is according to [30,31].

**Figure 2 polymers-12-00815-f002:**
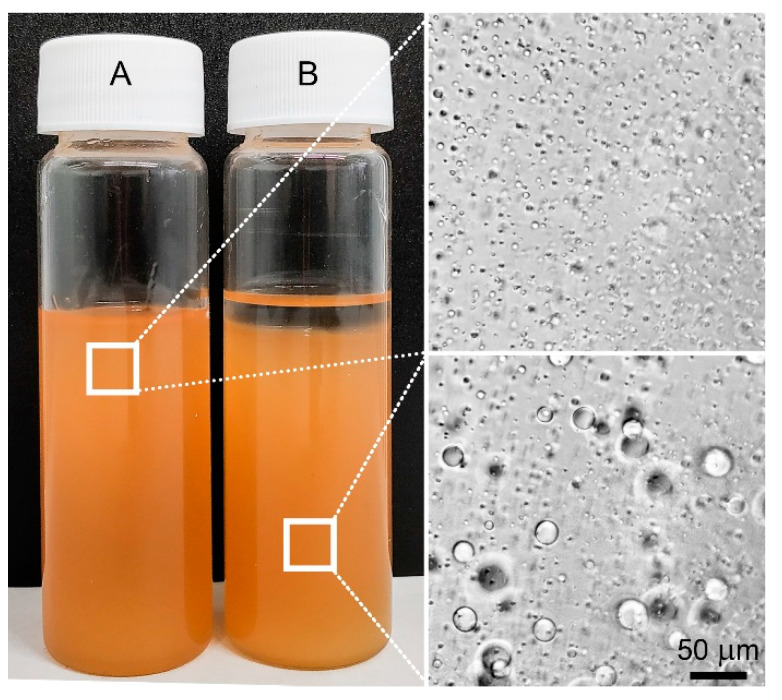
Polyacrylonitrile (PAN) solutions with the addition of 2 (**A**) and 10% of HMDSO (**B**).

**Figure 3 polymers-12-00815-f003:**
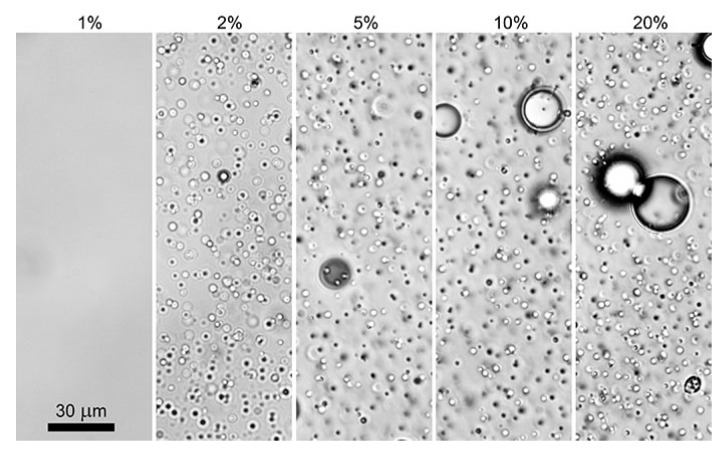
The morphology of mixtures of PAN solutions containing various amounts of HMDSO.

**Figure 4 polymers-12-00815-f004:**
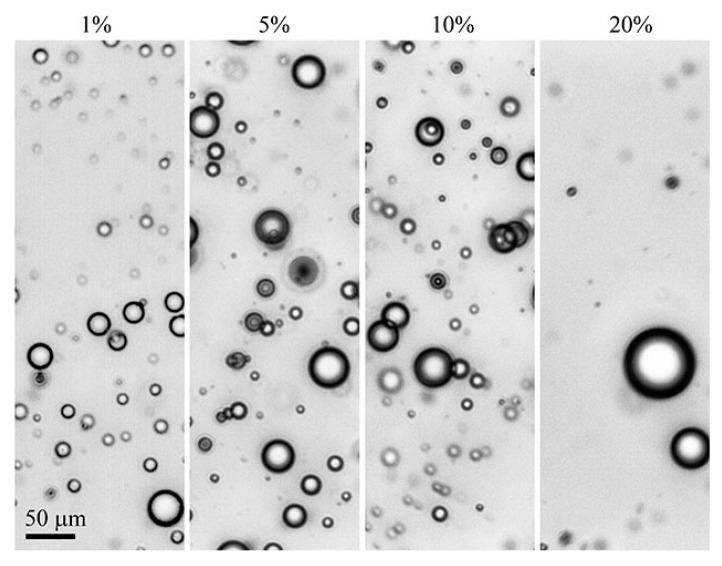
The morphology of a 20% PAN solution containing a different amount of PDMS-5.

**Figure 5 polymers-12-00815-f005:**
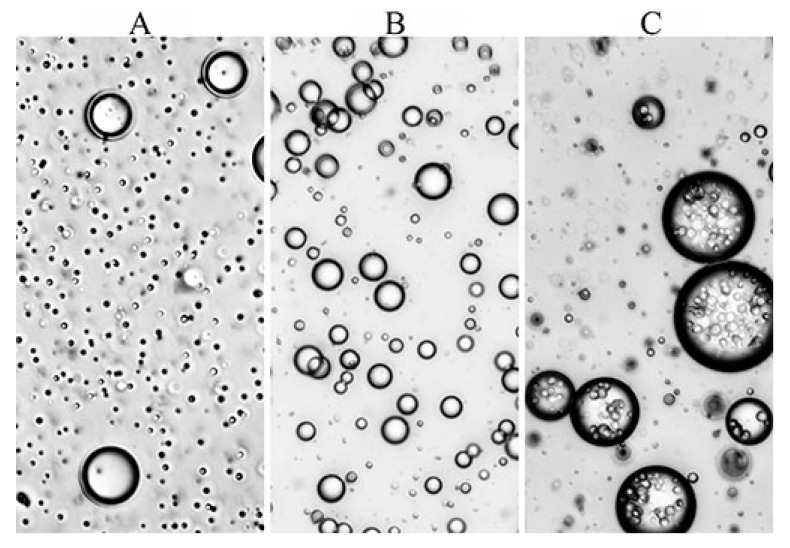
The morphology of a 20% PAN solution containing 5% HMDSO (**A**), PDMS-400 (**B**), or SKTN-E (**C**).

**Figure 6 polymers-12-00815-f006:**
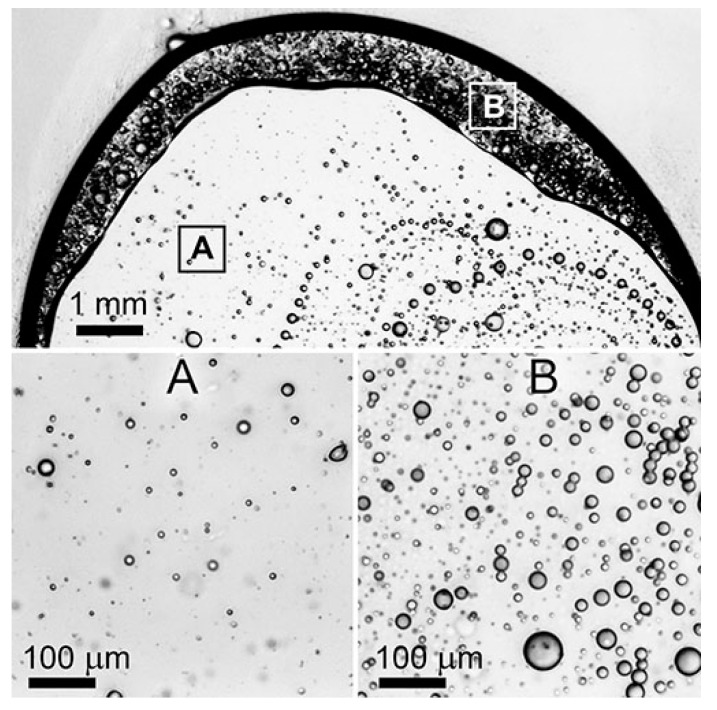
A micrograph of a drop of a PAN solution with 20% PDMS-400 in a narrow gap between two pieces of glass. The morphology along the radius is shown in inserts A (the middle part) and B (periphery).

**Figure 7 polymers-12-00815-f007:**
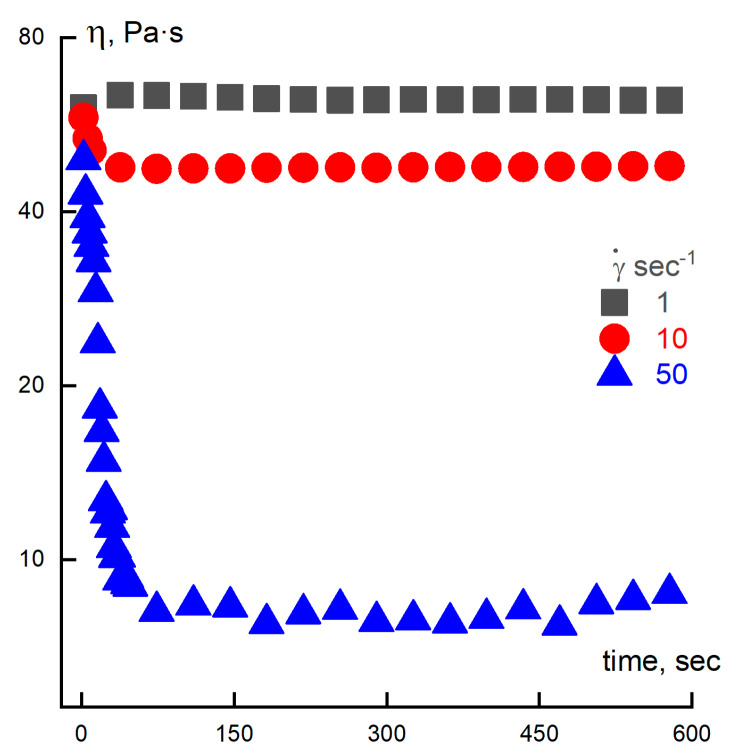
Evolution of the viscosity of the test emulsion with 5% of PDMS-100, measured at different shear rates.

**Figure 8 polymers-12-00815-f008:**
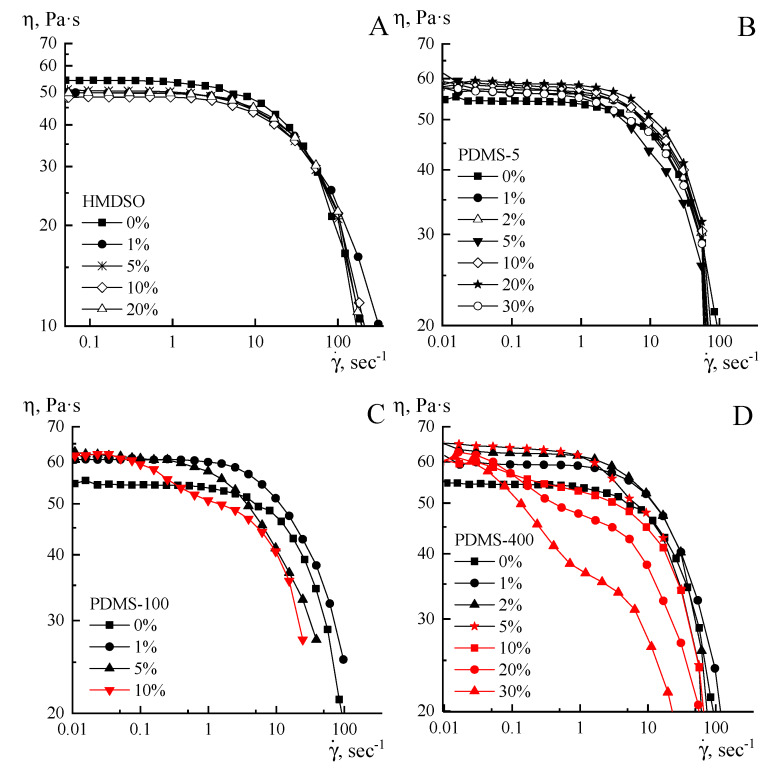
Flow curves of PAN solutions with various amounts of additives (indicated in the graphs) of polysiloxanes of various molecular weights: (**A**)—HMDSO, (**B**)—PDMS-5, (**C**)—PDMS-100, (**D**)— PDMS-400, and (**E**)—SKTN-E.

**Figure 9 polymers-12-00815-f009:**
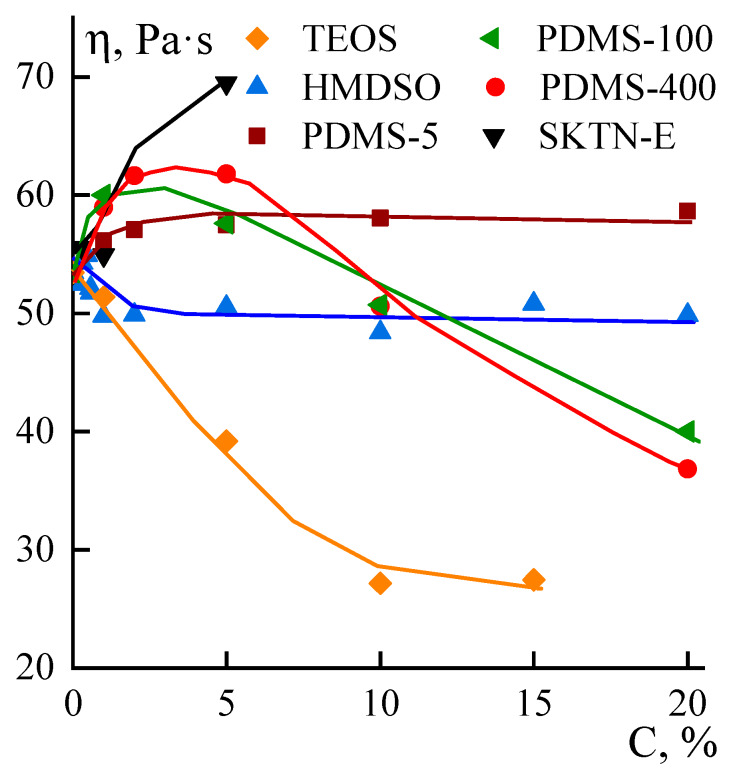
Dependences of viscosity, measured at a shear rate of 1 sec^−1^, of the compositions on the content of various PDMS.

**Figure 10 polymers-12-00815-f010:**
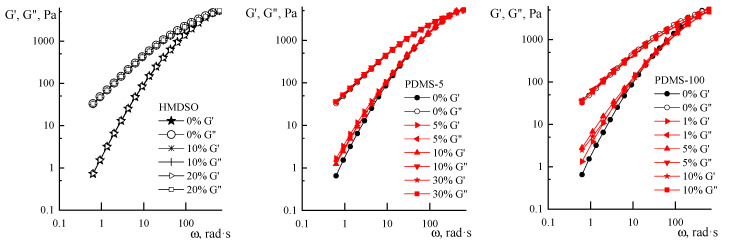
Frequency dependences of the elastic and loss moduli for PAN solutions containing PDMS.

**Figure 11 polymers-12-00815-f011:**
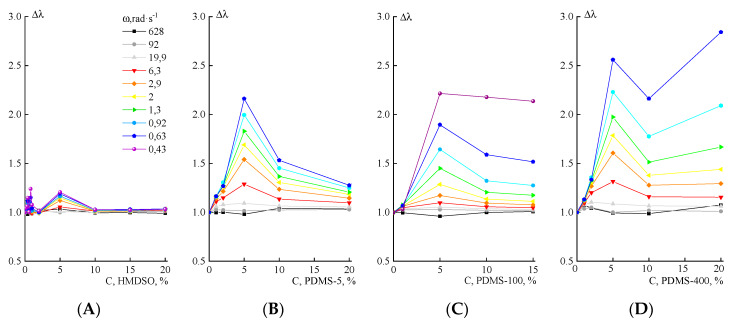
The dependence of the specific relaxation time at different frequencies on the concentration of polysiloxanes in the PAN solution: (**A**)—HMDSO, (**B**)—PDMS-5, (**C**)—PDMS-100, (**D**)—PDMS-400, and (**E**)—SKTN-E.

**Figure 12 polymers-12-00815-f012:**
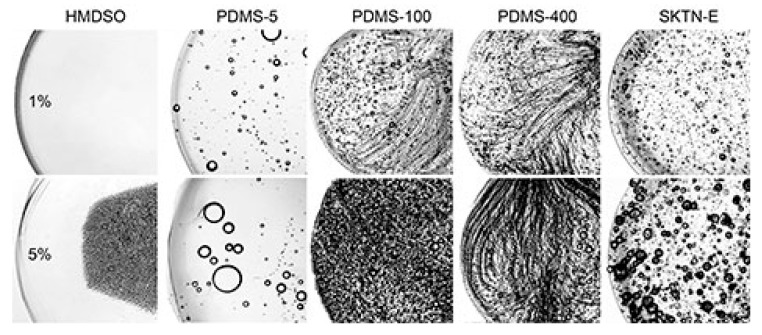
The morphology of heterophase droplets of compositions based on PAN solutions, containing 1% and 5% of HMDSO, PDMS-100, PDMS-400, and SKTN-E after the coagulation process.

**Figure 13 polymers-12-00815-f013:**
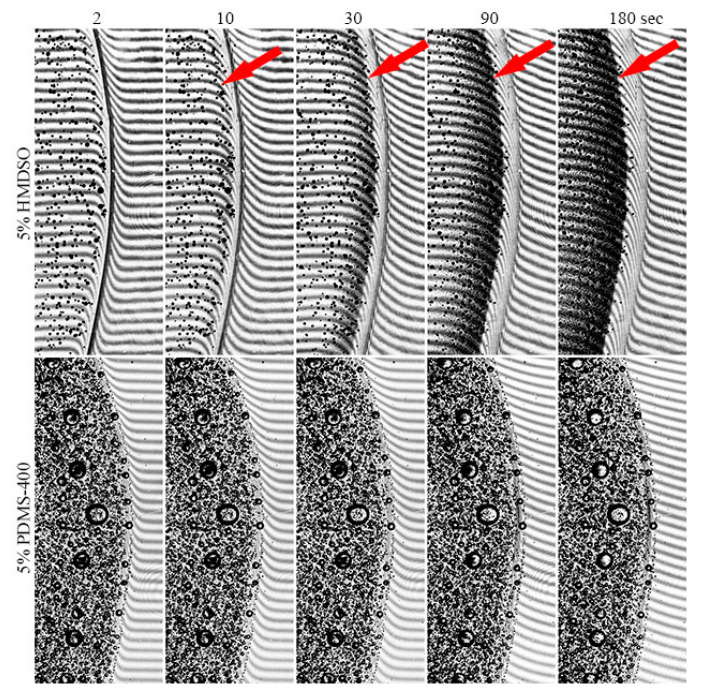
The evolution of interferograms upon the contact of PAN solutions containing 5% of HMDSO and PDMS-400 (left) with a coagulant (85% DMSO solution in water) (right) in time.

**Figure 14 polymers-12-00815-f014:**
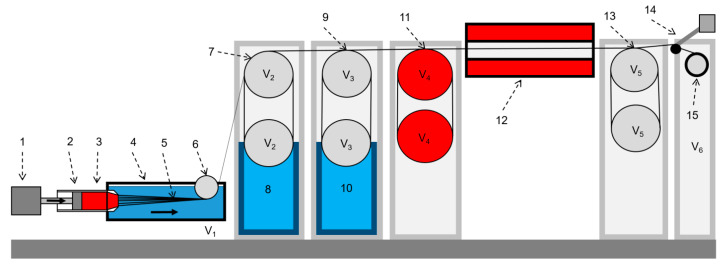
A scheme of the wet spinning stand. **1**—solution supply device; **2**—a syringe; **3**—dope; **4**—coagulant; **5**—complex yarn; **6**—take-up roller (spinbond hood); **7**, **9**—flushing rollers; **8**, **10**—washing bath; **11**—drying drums; **12**—zone of thermal extension; **13**—supporting rollers; **14**—yarn spreader; **15**—winding roller.

**Figure 15 polymers-12-00815-f015:**
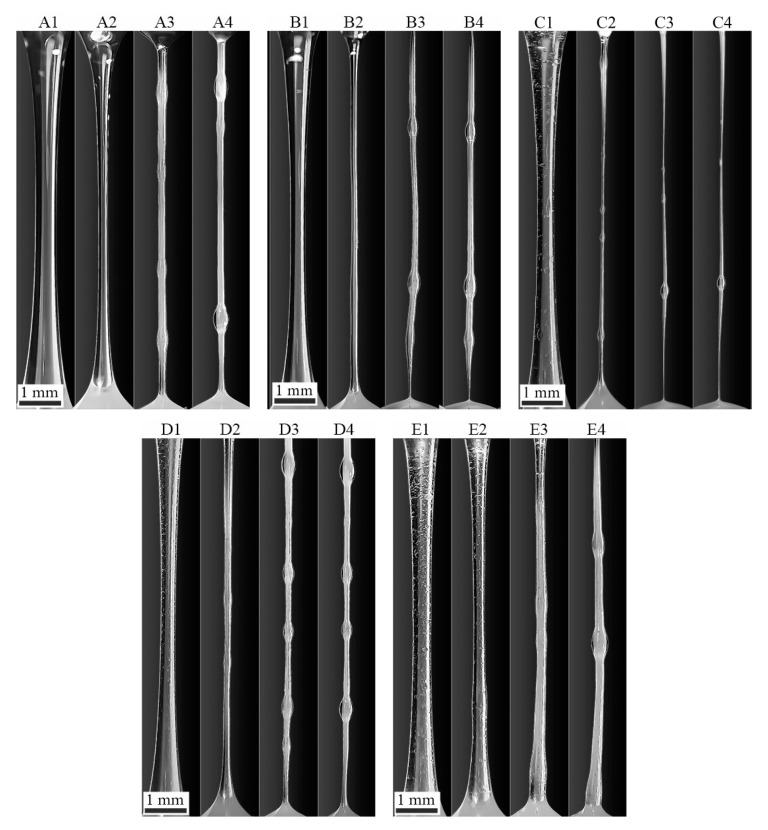
The evolution of the shape of the stretched jets based on 25% PAN solutions with polyorganosiloxane additives and the intensity of solvent release from jets. **1**—the initial moment of time, **2**—10 seconds, **3**—30 seconds, **4**—80 seconds after stretching the drop to a constant length. Additives: **A**—the neat solution; **B**—1% HMDSO; **C**—5% HMDSO; **D**—1% PDMS-100; and **E**—5% PDMS-100.

**Figure 16 polymers-12-00815-f016:**
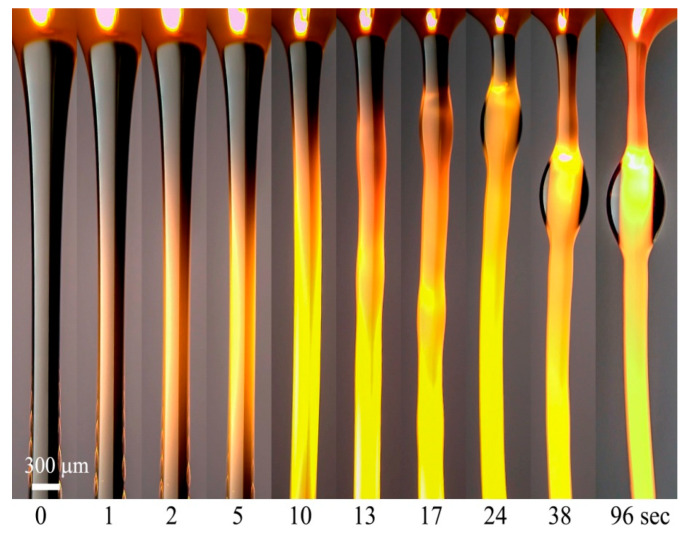
Phase separation at a constant draw ratio of PAN solution jet occurring in time (according to [31]).

**Table 1 polymers-12-00815-t001:** The characteristics of the polydimethylsiloxanes under consideration [29].

HMDSO	PDMS-5	PDMS-100	PDMS-400	SKTN-E
Density, g/sm^3^	0.79	0.913	0.966	0.970
Viscosity, Pa·s	0.5·10^−3^	5.5·10^−3^	9.6·10^−2^	0.39
M_n_, g/mole	162	384	~7000	~28,000

**Table 2 polymers-12-00815-t002:** Compositions under consideration: S—solution; E—emulsion; DE—double emulsion; MSE—macro-phase separation.

Concentration of Additive, %	Additive				
	HDMSO	PDMS-5	PDMS-100	PDMS-400	SKTN-E
1	S^1^	E	E^3^	E^3^	E^4^
2	E^1^ E^1^	E^2^	E^3^	E^3^	-
5		E^2^	E^3^	E^3^	DE^4^
10	MSE	MSE	E^2^	E^2^	-
20	MSE	MSE	MSE	E^2^	-

Emulsion stability: ^1^ ~ 2 days; ^2^ ~ 3 days; ^3^ ~ 2 weeks; ^4^ – more than a year.

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
