# Peer review of "Compositions Based on PAN Solutions Containing Polydimethylsiloxane Additives: Morphology, Rheology, and Fiber Spinning"

_polymers, 2020, doi:10.3390/polym12040815_

Round 1

Reviewer 1 Report

The paper « Compositions based on PAN solutions containing polydimethylsiloxanes additives: morphology, rheology, fiber spinning” by Kulichikhin et al. investigates the effect of several PDMS with various molecular weights on the morphology and rheological behavior of PAN solutions in DMSO.

The paper is clear and the conclusions are supported by the results. However, the manuscript must be checked by a native English speaker. Also, the authors must answer to the following questions:

  1. Page 3: line 100: which agitation regime was used?!
  2. Fig 10 and 11. Which technique was used for these figures?!
  3. The authors must discuss the stabilization mechanism of the emulsions and also to add some data about the static and dynamic stability of emulsions. For the dynamic stability it is possible to carry out some rheology tests at a constant shear rate as a function of time. See: https://doi.org/10.1016/j.colsurfa.2014.10.026
  4. Also concerning the rheology tests the authors should have used the following procedure: prior to all measurements, the samples must be treated at a shear rate of 50/s for 30 s and equilibrated for 2 min at 20 °C in order to standardize their history.
  5. Line 258-259: The authors state that the structure and dimensions of the interface affect the emulsions. In this context, the authors must explain the structure of the interface and how they measure this interface.
  6. The coalescence of the emulsions appears after which time?! If the authors observed a coalescence, then it means that the emulsions were destroyed.

Minor corrections:

  1. Line 20 : ratio of the disperse phase and the continuous medium…
  2. Page 1: lines 33-39: some review references need to be added
  3. Page 4, line: 139: the authors must indicate exactly which apparatus was used and not only to indicate the reference
  4. Page8, lines 261-264: the sentence: “if the stage…in the volume.” is not very clear.
  5. Shell-core or skin-core fibers?! The authors must be homogeneous within all the paper

In view of the above, I recommend the publication of this paper in Polymers only after major revisions.

Author Response

Please, see attachment. 

Reviewer 2 Report

The work by Kulichikhin et al. presents an interesting analyisis analysis of the compatibilization of polyacrylonitrile and polydimethylsiloxanes. The work is systematic and the results are physically sound.

My very minor comment is that authors should include a brief summary of the different samples studies and their type. A Table including the compositios will be useful.

Round 2

Reviewer 1 Report

The authors have made all the suggested corrections and therefore the manuscript can be published as it is.